# Exploring the Seasonal Dynamics and Molecular Mechanism of Wood Formation in Gymnosperm Trees

**DOI:** 10.3390/ijms24108624

**Published:** 2023-05-11

**Authors:** Thi Thu Tram Nguyen, Eun-Kyung Bae, Thi Ngoc Anh Tran, Hyoshin Lee, Jae-Heung Ko

**Affiliations:** 1Department of Plant & Environmental New Resources, Kyung Hee University, Yongin 17104, Republic of Korea; nguyenthutram1991@khu.ac.kr (T.T.T.N.); tranngocanh@khu.ac.kr (T.N.A.T.); 2Forest Bioresources Department, National Institute of Forest Science, Suwon 16631, Republic of Korea; baeek@korea.kr

**Keywords:** conifer, environment, epigenetic, genetic, gymnosperm, season, wood formation

## Abstract

Forests, comprising 31% of the Earth’s surface, play pivotal roles in regulating the carbon, water, and energy cycles. Despite being far less diverse than angiosperms, gymnosperms account for over 50% of the global woody biomass production. To sustain growth and development, gymnosperms have evolved the capacity to sense and respond to cyclical environmental signals, such as changes in photoperiod and seasonal temperature, which initiate growth (spring and summer) and dormancy (fall and winter). Cambium, the lateral meristem responsible for wood formation, is reactivated through a complex interplay among hormonal, genetic, and epigenetic factors. Temperature signals perceived in early spring induce the synthesis of several phytohormones, including auxins, cytokinins, and gibberellins, which in turn reactivate cambium cells. Additionally, microRNA-mediated genetic and epigenetic pathways modulate cambial function. As a result, the cambium becomes active during the summer, resulting in active secondary xylem (i.e., wood) production, and starts to become inactive in autumn. This review summarizes and discusses recent findings regarding the climatic, hormonal, genetic, and epigenetic regulation of wood formation in gymnosperm trees (i.e., conifers) in response to seasonal changes.

## 1. Introduction

Covering 31% of the world’s land area, forests play important ecological and economic roles by regulating the world’s carbon, water, and energy cycles [1,2]. Trees have accumulated approximately one-third of atmospheric carbon dioxide over the past few decades through the tightly regulated annual growth, thereby acting as significant long-term global biotic carbon sinks [3]. The resulting biomass can serve as a source of lignocellulose to produce biomaterials, and has recently been used as a raw material to produce liquid biofuels and other value-added products [4]. Conifers are the most diverse and common group of gymnosperms, with about 615 species distributed worldwide [5]. Despite being outnumbered by angiosperms, gymnosperms contribute significantly to woody biomass formation, accounting for over 50% of the world’s timber production [5,6].

Wood (i.e., secondary xylem) is composed primarily of cellulose, xylan, and lignin. Wood formation begins with the cell division of the vascular cambium into secondary xylem inside and secondary phloem outside [7]. Next, xylem daughter cells expand to their final shape and size before secondary cell wall (SCW) deposition and programmed cell death (PCD) occur [7,8]. Wood formation in trees is a cyclical process that is strongly influenced by seasonal changes in environmental conditions, such as day length and temperature. Trees must be able to detect and respond to these signals in order to regulate their activity, with growth occurring during the growing season when conditions are optimal (spring and summer) and dormancy during times when conditions are less favorable for growth (fall and winter). In the active growing season, annual growth rings of wood are formed due to the production of earlywood (i.e., springwood) inside and latewood (i.e., summerwood) outside [9]. The molecular regulation of wood formation and seasonal control in trees, including conifers, has been explored using multiomic approaches and a phytohormonal perspective [10,11,12,13].

Non-coding RNAs are involved in the epigenetic regulation of wood formation [14,15,16]. MicroRNAs (miRNAs), which are 20 to 24 nucleotides in length, are widely recognized as one of the major epigenetic regulating factors [17,18]. miRNAs change the level of proteins by regulating the mRNA expression of target proteins without modifying the sequences of the genes, and epigenetic modifications such as DNA methylation and histone modifications can further influence miRNA expression [18]. In addition to negatively regulating target genes, miRNAs can also positively regulate target genes [17].

This review summarizes and discusses recent findings on the climatic, hormonal, genetic, and epigenetic regulation of wood formation in gymnosperms, especially conifers, during seasonal changes.

## 2. How Climate Affects Wood Formation

Climate conditions are the first of many variables that influence how trees produce wood, resulting in the annual wood-forming rhythm [9]. Temperature and photoperiod influence cambium reactivation and secondary xylem production [19,20]. In Japanese cedar (*Cryptomeria japonica*), the shoot and young needle sense temperature and photoperiodic signal and show seasonal transcriptome changes [21]. Photosynthesis produces C compounds for cell wall formation, and rainfall supplies water to create the right turgor pressure on the cell wall, which results in cell expansion [22,23].

Warmer temperatures after a cold winter are thought to be one of the main factors in determining when cambium resumes at the start of the growing season (early spring) [13,19,24,25]. The earliest cambium activity of the year is recorded at 10 °C in Chinese red pine (*Pinus massoniana* Lamb.) and Scots pine (*Pinus sylvestris*), and 5.6–9 °C for European larch (*Larix decidua*), Swiss pine (*Pinus cembra*), Norway spruce (*Picea abies*), Balsam fir (*Abies balsamea*), Bosnian pine (*Pinus leucodermis*), and Mountain pine (*Pinus uncinate*) [19,25,26]. Threshold temperature depends on the species and age [24,25]. Three to four weeks after cambium reactivation, cambium cells are actively dividing, and xylem differentiation occurs through cell expansion, SCW formation, and PCD [7,25]. Temperatures favorable for cambium activity are about 17 °C for Scots pine and about 25 °C for Chinese red pine [19,26]. When the air temperature decreases in the fall (about 15 °C for Chinese red pine and 8–10 °C for *Chamaecyparis obtuse*), cambium enters dormancy and the wood formation process ceases [19,24].

During the long days of spring and summer, photosynthesis is active, providing a constant source of carbon for cell wall formation [19,21,23]. Prolonged photosynthesis provides sufficient ingredients for cell wall formation during the beginning of the active season, when cambium cells divide rapidly and xylem cells expand as much [20,26]. As a result, earlywood is formed, which has a large area, thin cell wall, and brilliant color [26]. The constant supply of sucrose produced from starch reserves may promote cambium cell division and xylem formation [19]. The inducing soluble sugar that changes the osmotic pressure may contribute to turgor pressure necessary for cell expansion, so it parallel with the activity of cambium cell [24,26]. During mid to late summer (July to September) in the Northern Hemisphere, day length is very long. It may suggest that the abundant source of carbon at this time provides for cells in SCW formation progress, leading to thick cell-walled latewood. At the half end of the active season, when conditions are no longer favorable, cambium activity decreases and xylem cells do not expand [24,26].

Precipitation strongly influences cambium activity and xylem differentiation because they require turgor pressure during cell expansion [19,27]. In the beginning of the growing season, abundant rainfall increases cambium division, resulting in the formation of many layers of xylem cells with a large cell area called earlywood. Conversely, in the second half of the growing season, drought reduces cambium division, resulting in the formation of a few layers of xylem cells with a small cell area called latewood [26,27]. Latewood formation is strictly influenced by environmental conditions [24]. Cambium is known to rest in the cold weather of fall [24,26]. In fact, though, drought is considered to be the major factor that leads to cambium inactivation. Growth is enhanced and the latewood ring is thin under ideal conditions, whereas latewood formation is high under low precipitation and low temperature in Douglas-fir (*Pseudotsuga menziesii*) [28]. In Scots pine, cambium rest and latewood forms in the hottest time of summer (around 27 °C), under low precipitation, due to the limitation of cell expansion [26]. However, cambium division, cell expansion, and SCW formation have two peaks in the subtropical forest: one during the rainy season (middle of April) and another during the dry season (early October, by the time of long sunshine duration) [19].

Dormancy is crucial for the growth of wood since it enables the tree to preserve energy and resources during the winter. When a tree goes dormant, it diverts energy and nutrients from growth activities to other survival-related processes. This enables the plants to shield the meristems from unfavorable environmental circumstances and coordinate the time of their growth cycle for favorable environmental factors [29]. Cold temperature is thought to be the key factor in dormancy induction. In Hinoki cypress (*Chamaecyparis obtusa*), a decrease of temperature from 25 °C to 8–10 °C inhibits cambium cell division [24]. However, this effect is still not well understood.

At the beginning of dormancy, the ‘rest’ phase (endodormancy or physiological dormancy) is the time after the cambium becomes inactive, which takes 2–4 weeks and is influenced by endogenous factors, so the cambium is unable to divide even under favorable conditions. The ‘quiescence’ phase (ecodormancy or environment dormancy), on the other hand, occurs after that and is the result of environmental factors, so it can stop when climatic conditions become favorable [19,24,25,29]. The difference between the rest and quiescence was shown in the study of vacuoles in Lodgepole pine (*Pinus contorta*). Cambium cells in the rest phase have many small and elongated vacuoles, whereas cambium cells in the quiescence phase have fewer rounded big vacuoles [29]. Vacuoles play an important role in the storage of reserves in cell activity. Thus, the change in vacuole status relates to the change in cell activation [29]. Until now, the genetic and hormone regulation in the transition from rest to quiescence is still unknown in gymnosperm, but chilling in the winter is required for the onset of quiescence [29]. In angiosperm, it has been suggested that, after 4 weeks of short-day treatment, cambium stops dividing and goes to the quiescence phase with the decline of histone H1 kinase activity in A- and B-type cyclin kinase (CDKA/B) kinase complex, which regulates cell cycle. However, longer short-day treatment (6 weeks) leads to the rest phase, which reduces and inhibits CDKA/B, followed by the reduction of retinoblastoma (Rb) phosphorylation activity from CDKA protein complex. Thus, G1 to S phase in cell cycle is blocked [30].

In summary, the transition from active growth to rest phase of dormancy can be caused by a decrease in day length, precipitation, and temperature. The transition from rest to quiescence phase is affected by chilling, which leads to structural, histochemical, and functional changes in cambial cells. The transition from quiescence to reactivation phase depends on temperature and day length.

## 3. Hormonal Regulation of Seasonal Wood Formation

Since the 1960s, it has been hypothesized that hormones influence wood formation [31]. Cells in the vascular cambium are constantly forced to decide between differentiation and maintenance. The crucial first step in the differentiation of cambium cells into xylem cells is mediated by hormone signaling, particularly auxin signaling [32].

Auxins have been demonstrated to induce extracellular acidification, which is a necessary factor for the relaxation of cell walls and subsequent cell expansion [33,34]. Auxins also promote cell proliferation by upregulating the cyclin genes required for the G1 to S phase transition [33]. In Chinese red pine, auxin accumulation was found to be initial step in the cambium reactivation process [19]. Indole-3-acetic acid (IAA), an endogenous auxin, is produced in young buds and leaves, and can be polarly translocated down to the stem, where it induces cambium cell differentiation [19,24]. Auxins are also produced in the cambium zone of the stem and polarly transported to developing xylem, where auxin synthesis is low. Therefore, auxin concentrations are the highest in the cambial zone and gradually decline toward the xylem [12,35]. Polar auxin transport requires active transporters. Genes responsible for polar auxin transport such as PIN-like auxin efflux carriers and auxin-induced proteins are specifically expressed during the reactivation and active stages of cambium differentiation in addition to the production of earlywood and latewood [36,37]. Auxin signaling genes such as *Auxin Response Factors* (*ARFs*) were also found to be induced in the active cambium of Chinese fir (*Cunninghamia lanceolata*) [36]. Polar auxin transport and auxin signaling genes are therefore regulated by season [36]. The concentration of IAA is high during the cambium active stage, peaks in mid-spring (e.g., April) in parallel with cell enlargement, cell division, and cell wall thickening, and then decreases in early summer (e.g., June) [26].

Under favorable conditions in the active growing season, a significant number of xylem cells known as earlywood (i.e., spring wood) form rapidly. When conditions are no longer favorable at the end of the active season, xylem forms slowly with a small number of cells that are small and have thick cell walls and dark coloration (latewood). This occurs annually, resulting in annual rings containing earlywood (inside) and latewood (outside) [38]. Fajstavr et al. found that when the IAA level peaked, a large number of tracheids with a large area and thin cell wall were formed in earlywood [26]. Early studies showed that exogenous treatment with IAA rapidly increased the formation of earlywood tracheids in the cambia of Red pine (*Pinus resinosa*) [31]. Higher auxin concentrations in Chinese red pine triggered an increase in xylem cell production [27]. Unlike earlywood, the influence of auxins in the formation of latewood is still unclear. For example, auxin levels in the cambium of Balsam fir and Korean red pine (*Pinus densiflora*) decreased during the early- to latewood transition, whereas there was no significant change in auxin levels in Scots pine [26,39].

Cytokinin biosynthesis genes have been reported to enhance cambium cell division [40]. The total amount of endogenous cytokinins in mid-summer is higher than that in mid-winter. Isopentenyl-type cytokinins are signals for cambium cell division in Larch (*Larix kaempferi*) before the active biosynthesis of endogenous IAA. Isopentenyladenine (iP) appears to be an active form produced during the active season, while isopentenyladenosine (iPA, the 9-riboside of iP) appears to be a storage form that is converted to iP [11].

Gibberellins (GAs) have been reported to promote xylem cell differentiation and elongation in *Arabidopsis* and *Populus* [41,42]. Exogenous GA induces dramatic cambium proliferation [42]. Overexpression of *GA20-oxidase1*, a rate-limiting enzyme in GA biosynthesis from Korean red pine, leads to gelatinous fiber development in transgenic hybrid poplars [43]. Moreover, GA promotes both cell proliferation and differentiation through inhibiting DELLA, which limits cell production and elongation [44]. GA has critical interaction with auxin through induction of *PIN1*, leading to the stimulation of cambium proliferation [45]. GAs have been reported to be key signals for the release of dormancy in gymnosperm seeds and buds [46,47,48]. In Manchurian red pine (*Pinus tabuliformis*), GAs are induced in the early spring to break bud dormancy by blocking DELLA, resulting in expression of *FTL2* (*Flowering Locus T/Terminal Flower1-like 2*) that then induces flowering [48]. The induced expression of *GA20-oxidase1* and *GA20-oxidase2* genes increased GA concentrations in Ginkgo (*Ginkgo biloba*) embryos upon the dormancy release of seeds [47].

Abscisic acid (ABA) has been shown to be an inhibitor of cambium activity. ABA synthesis was dramatically reduced in the beginning of active season by MYB transcription factors and miRNAs, leading to the release of cambium cells from dormancy [49]. However, Funada et al. found no changes in endogenous ABA levels between the active and resting stages of cambium in either Korean red pine or larch [11,50].

Collectively, the available evidence suggests that an increase in GAs and decrease in ABA releases cambium cells from their dormancy, while cytokinins initiate cell division in the cambium. Auxins are involved in cell division and elongation in both the reactive and active stages of cambium differentiation.

## 4. Genetic Regulation of Seasonal Wood Formation

Genetic control of seasonal aspects of wood formation has been studied by RNA-sequencing and genome-wide association studies [10,13,36]. The key gene in xylem development and secondary cell wall deposition was examined in relation to the seasonal feature. Warm spring air is critical to break dormancy, leading to cambium cell reactivation and division, including cell cycle gene activity [24]. The expression of three CDKB proteins, which are involved in the G2 to M transition (entry into division) in the cell cycle, was shown to peak at the end of spring and decline after cambial growth stopped [13]. Larch *LaCYCB1;1* and *LaCDKB1;3* were abundant at the start of the active season, and *LaMYB20*, *LaRAV1*, *LaCYCB1;1*, and *LaCDKB1;3* were expressed even as the temperature increased. LaMYB20 promotes expression of *LaCYCB1;1*, while LaRAV1 promotes expression of *LaCDKB1;3*. The LaCYCB1;1 interacts with LaCDKB1;3, promoting cambium cell division and cell cycle resumption [51]. In Chinese fir, levels of cyclin A/B and histone H4, markers of cell division and DNA replication, respectively, also increased in the active growing season and decreased in the dormant season [36,52]. Suppression of the abiotic stress response in early spring was hypothesized to trigger the active formation of wood through the inhibition of *Leaf Curling Responsiveness* (*LCR*) and *MYB5*, which are involved in salt stress and heat stress response, respectively [49,53,54]. Cell expansion is the first step of xylem cell differentiation [7]. In Norway spruce, cambium reactivation begins with the expression of alpha-*Expansins* (*EXPAs*), a marker of cell wall expansion [13,55]. The cell wall loosens and elongates during cell expansion due to the elevated expression of genes encoding xyloglucan endotransglycosylases (XTH), pectin methylesterase, pectin esterase, pectate lyase, expansin, beta-1,3-glucanase, and endo-1,4-beta-glucanase [36].

SCW formation includes coordinated transcriptional regulation of genes involved in cellulose, hemicellulose, and lignin biosynthesis, which requires a multifaceted and multilayered transcriptional regulatory program that is active in the growing season [7,13,36,56,57,58]. MYB26 regulates *NST1* and *NST2*, which are responsible for SCW formation. MYB26 also directly activates genes involved in cellulose and lignin biosynthesis [59]. NST1/2 and NST3 (also known as SND1) regulate SCW thickening of xylem fibers [60,61]. Some members of the NAC family control the formation of tracheids in Korean red pine [57]. MYB46 and MYB83, master switches of SCW formation, coordinately regulate many downstream transcription factors (e.g., MYB58/63/85 promote lignin biosynthesis; MYB4/7/32/75 and KNAT7 inhibit lignin and cellulose biosynthesis; SND2/3 and MYB20/42/43/52/54/69/103 promote cellulose, hemicellulose, and lignin biosynthesis) [58,59]. The transcription factors AS2, MYB16, and MYB20 were predicted to be involved in the seasonal regulation of SCW formation based on their expression patterns [13]. The lignin polymerization gene, *Laccase*, is expressed only during the active growing season, but monolignol biosynthesis can occur during the cambial dormancy phase and reaches a second peak in mid-winter [13]. This demonstrates that activation of monolignol biosynthetic genes (*CAD*, *COMT*, *CCoAOMT*, etc.) during the winter helps to provide defense against stress caused by freezing or UV-B radiation [13].

Earlywood is formed between spring and early summer (e.g., April and June), whereas latewood is formed in late summer (e.g., July and August) [26]. Earlywood tracheids have a wider lumen diameter than latewood, and therefore receive 90% of conducted water compared to the 10% that latewood receives [38,62]. In Norway spruce, NAC transcription factors and cellulose synthases were found to be abundant during the transition from earlywood to latewood [10]. In latewood formation, RNA-dependent RNA polymerase (RDR), which is involved in posttranscriptional gene silencing, is expressed at high levels [13]. RDR2 interacts physically with DNA-dependent nuclear RNA polymerase IV (Pol IV) and somehow engages with Pol IV transcripts, allowing them to produce double-stranded RNAs. Double-stranded RNAs were diced and methylated, then undergo DNA methylation and gene silencing [63]. The mechanisms regulating the earlywood to latewood transition are still under investigation.

Cambium cells decline to divide around the middle of summer and enter a resting phase at the end of summer due to suppression of A-type and B-type cyclin genes [13,26]. Xylem cells subsequently stop expanding in the middle of summer, and SCW formation finishes before the beginning of fall [26].

PCD is the final step of the xylem differentiation when vacuoles collapse by tonoplast rupture and digestive enzymes are released to degrade cell components [7]. Tracheid cells of Norway spruce undergo PCD 1–3 months after their formation [64]. PCD is induced by many proteinases and RNases including Xylem Cysteine Protease 1 (XCP1) and XCP2, MetaCaspase 9 (MC9), Bifunctional Nuclease1 (BFN1), Acualis5 (ACL5), Responsive to Dehydration 21A proteinase (RD21A), Cysteine Endopeptidases (CEPs), Xylem Bark Cysteine Peptidase3 (XBCP3), and class I RNAse (RNSI). All these proteins are upregulated during the earlywood PCD process [13]. NAC transcription factors regulate both xylem formation and tracheid PCD in Korean red pine [57]. PCD of earlywood tracheids begins in the middle of the active season, whereas a few latewood cells still survive with intact vacuoles and possess cellular contents at the end of winter (Eastern white pine; *Pinus strobus*) and even at the beginning of the next active season (Norway spruce) [13,65]. Induction of *Terminal flower* (*TFL*, involved in growth cessation) and *β-amylase* (*BAM3*, involved in cold response) in latewood are signals that the woody tissue is transitioning into dormancy [10,66]. Expression of *Flowering Locus T* (*FT*)/*TFL1-like2* (*PaFTL2*) peaked at the beginning of the fall [13].

## 5. Epigenetic Regulation of Seasonal Wood Formation

The miRNAs have been widely studied in gymnosperms through small RNA sequencing technology [67,68,69]. Many miRNAs that regulate cambium activity in forest trees have been identified [49,70,71]. The miR397a downregulates the *laccase* gene in poplar (*Populus trichocarpa*) [72]. miR319 downregulates *TCP4*, leading to an increase in SCW in inflorescence stems of Arabidopsis [73]. miR166 promotes vascular differentiation by inhibiting the class III HD-ZIP family transcription factor gene *PtaHB1* in *Populus* [74]. miR165 induces SCW in pith by downregulating *AtHB15* in Arabidopsis [75]. The miRNA transcriptional profile of xylem during different stages of SCW formation has been studied in Chinese red pine [15]. Like hormonal and genetic regulation, regulation by miRNAs is also seasonal, as first reported by Ko et al. [74]. The activity of miRNAs in Chinese cedar and Ginkgo has been well studied in the cambium during the reactivation, activation, and dormancy stages [46,49].

Early in the cambium reactivation, hormone responses are modulated by an abundance of miR166, miR159, and miR160 in Chinese cedar and Ginkgo [46,49]. miR166 inhibits ABA signaling in plants and is expressed at low levels during somatic embryo maturation [76,77]. miR159 suppresses GA signaling by inhibiting MYB33 and MYB101, and renders plants hyposensitive to ABA [78]. In addition, miR159 is induced by GA signaling, and serves as a feedback inhibitor that limits the GA signaling response through GAMYB [79,80]. miR160 induces both formation and transport of IAA [81]. Both miR166 and miR160 were upregulated in the transition from reactivation to activation of the cambium, and this was associated with reduced ABA but increased IAA content in Chinese cedar [49].

To break dormancy, stress resistance responses are suppressed by upregulation of miR394c, miR858, and a novel microRNA cln-miR08 [49,71]. miR394 targets *LCR*, a F-BOX family protein involved in salt and drought stress responses [54]. miR858e targets *MYB5* involved in heat stress [53]. cln-miR08 targets the *NBS-LRR* resistance gene involved in biotic stress [71,82]. In cambium reactivation, miR156 is induced to suppress *SQUAMOSA promoter binding protein-like* (*SPL*) gene, which encodes a protein that regulates the juvenile-to-adult transition, eventually reducing the expression of *miR172*. In contrast, in the cambium activation stage, a decrease in *miR156* expression leads to an increase in *miR172* expression, which promotes vegetative induction [71,83]. miR397b also suppresses *LAC4* and *LAC11*, which play important roles in lignin polymerization [49]. A decrease in lignin content can increase cell wall permeability, which helps in the expansion of cambium cells by increasing osmotic penetration [84].

In the activation stage, when cambium division is rapid, miR166a, an inhibitor of the negative regulator of cambium differentiation, HB-15, in both Chinese fir and Chinese cedar, was found to be expressed at high levels [49,71,85]. miR164 and miR319 were also expressed at high levels in the active stage in Ginkgo and Chinese fir [46,71]. *AtNAC2*, which promotes cell death and senescence, is suppressed by *miR164* [86]. miR319 inhibits *TCP24*, a repressor of SCW formation [87].

At the end of the active stage, when the division of cambium cells slows down, *Growth Regulating Factor* (*GRF*) genes are suppressed to prepare for dormancy. miR396 was found to be upregulated and inhibit seven of nine *GRF* genes [49,88].

In the dormancy stage, miRNAs such as miR482, miR2118, miR5261, and miR2950 are expressed at high levels in Ginkgo, suggesting that these miRNAs are involved in the regulation of the cambial activity transition [46]. miR482 targets *Nucleotide Binding Site-Leucine-rich Repeat* (*NBS*-*LRR*) genes, which are present in immune system [89]. miR2118 targets genes responsive to fungal infection [90]. miR5261 is involved in the magnesium deficiency response by enhancing the activity of the antioxidant system, and is inhibited by high temperatures [91,92]. miR2950 enhances the plant immune system via a yet unknown mechanism [93]. In conclusion, miRNAs play a crucial role in regulating the different stages of cambium activity in gymnosperms, and their expression levels are tightly coordinated with hormonal and genetic regulation.

## 6. Conclusions and Future Perspectives

Seasonal regulation of wood formation is critical to conifer survival and growth. All genes and miRNAs involved in the dynamics of seasonal wood formation in gymnosperm trees described in this review were presented in Table 1 and Figure 1, and the seasonal dynamics of wood formation in gymnosperm trees were summarized in Figure 2. The initiation of wood formation in early spring is triggered by environmental factors such as increased atmospheric temperature and longer day length, which stimulate the emergence of new shoots and leaves. The increase in the synthesis of auxins in new shoots and leaves and their transport to the vascular cambium, or production of auxins in the cambium, leads to the reactivation of this tissue, a critical component of secondary growth in plants. This reactivation process is regulated by a complex interplay among various phytohormones, including GAs, cytokinin (iP), and IAA, as well as various genes and microRNAs such as miR166, miR160, and miR159. Proliferation of cambium cells and formation of xylem is facilitated by the coordinated regulation of these factors. The formation of earlywood and latewood has similarities and differences, and is subject to regulation by various hormones and genes. Cambial rest or the cessation of cambial activity is induced by the suppression of *GRF* genes by miR396, and is accompanied by the accumulation of factors promoting abiotic and biotic stress resistance as well as gene silencing through RNA-directed DNA methylation.

Overall, seasonal regulation of wood formation in conifers is a complex and multi-faceted process that involves the interplay of various hormones, genes, miRNAs, and environmental cues. However, the entire process is still not fully understood. Thus, the information presented in this review should serve as a valuable foundation for further study. Future research in this area could have significant implications for the sustainable production of wood and the conservation of forest ecosystems that are critical to mitigating the global climate crisis.

## Figures and Tables

**Figure 1 ijms-24-08624-f001:**
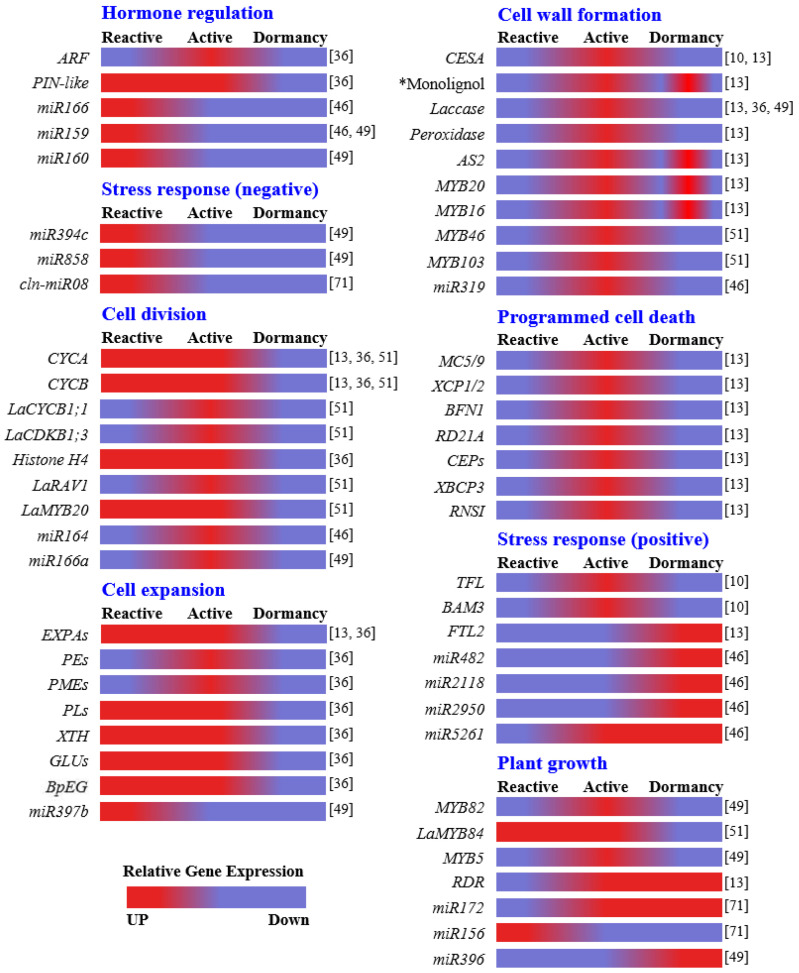
**The seasonal expression of genes and miRNAs involved in wood formation of gymnosperm trees.** The relative expression of each gene or miRNA is shown as a color bar according to reactive, active, and dormancy: red indicates upregulation and purple indicates downregulation (bottom left). * Monolignol biosynthesis genes are *CAD*, *COMT*, *CCoAOMT*, *CCR*, *C3H*, *C4H*, *4CL*, *CSE*, *PAL*, and *HCT*. Descriptions of each gene and miRNA are shown in Table 1. References are to the right of each color bar.

**Figure 2 ijms-24-08624-f002:**
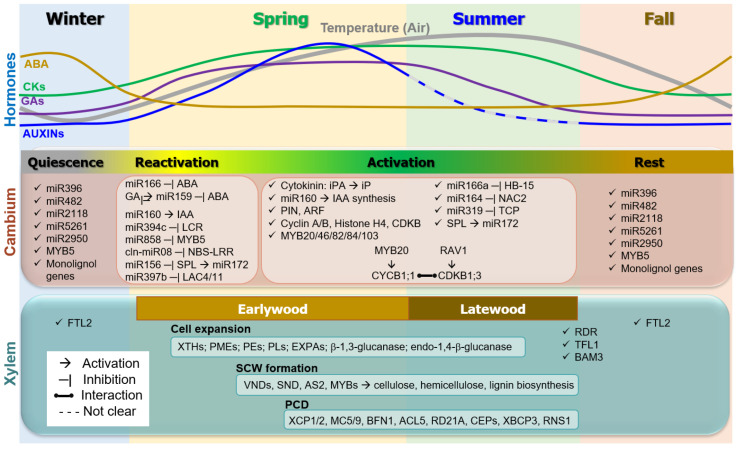
**Summary of seasonal dynamics of wood formation in gymnosperm trees.** Changes of cambium activity (i.e., quiescence, reactivation, activation, and rest) and xylem formation (i.e., cell expansion, SCW formation and PCD) with earlywood and latewood under the seasonal signaling were summarized. Seasonal changes of hormone concentrations (e.g., Auxins, cytokinins, ABA, and GAs) were shown with temperature fluctuation above. Representative genes and miRNAs responsible for each process were shown. Full gene names and references are in the main text.

**Table 1 ijms-24-08624-t001:** Recent studies have explored the dynamics of seasonal wood formation in gymnosperm trees.

Gene (miRNA)	Description (Function)	Studied Plant Species	Ref.
**Hormone regulation**			
*ARF*	Auxin response factor (responsive to auxin signaling)	*Cunninghamia lanceolata*	[36]
*PIN-like*	Auxin efflux carrier (exporting auxin from cell)	*Cunninghamia lanceolata*	[36]
*miR166*	(Inhibit ABA signaling)	*Ginkgo biloba* L.	[46]
*miR159*	(Make hyposensitive to ABA by inhibiting GA signaling)	*Cryptomeria fortunei* Hooibrenk, *Ginkgo biloba* L.	[46,49]
*miR160*	(Induce accumulation and transport of IAA)	*Cryptomeria fortunei* Hooibrenk	[49]
**Stress response (negative)**			
*miR394c*	(Inhibit *LCR*, involved in salt and drought stress)	*Cryptomeria fortunei* Hooibrenk	[49]
*miR858*	(Inhibit *MYB5*, involved in heat stress)	*Cryptomeria fortunei* Hooibrenk	[49]
*cln-miR08*	(Inhibit *NBS-LRR*, involved in biotic stress)	*Cunninghamia lanceolata*	[71]
**Cell division**			
*CYCA*	A type cyclin (required for the G1 to S phase transition)	*Cunninghamia lanceolata*, *Larix kaempferi*, *Picea abies*	[13,36,51]
*CYCB*	B type cyclin (required for the G2 to M phase transition)	*Cunninghamia lanceolata*, *Larix kaempferi*, *Picea abies*	[13,36,51]
*LaCYCB1;1*	(Required for G2 to M phase transition)	*Larix kaempferi*	[51]
*LaCDKB1;3*	Cyclin dependent kinase (Required for G2 to M phase transition)	*Larix kaempferi*	[51]
*Histone H4*	(Involved in DNA replication)	*Cunninghamia lanceolata*	[36]
*LaRAV1*	Homolog of poplar *RAV1* (induce *LaCDKB1;3*)	*Larix kaempferi*	[51]
*LaMYB20*	Homolog of *AtMYB20* (induce *LaCYCB1;1*)	*Larix kaempferi*	[51]
*miR164*	(Inhibit *NAC2*, promoting cell death and senescence)	*Ginkgo biloba* L.	[46]
*miR166a*	(Inhibit *HB-15*, negative regulator of cambium differentiation)	*Cryptomeria fortunei* Hooibrenk	[49]
**Cell expansion**			
*EXPAs*	*α Expansin* (break noncovalent bonds between cell wall components)	*Cunninghamia lanceolata*, *Picea abies*	[13,36]
*PEs*	*Pectin esterase* (catalyze the de-esterification of pectin into pectate and methanol)	*Cunninghamia lanceolata*	[36]
*PMEs*	*Pectin methylesterase* (catalyze the demethoxylation of pectin)	*Cunninghamia lanceolata*	[36]
*PLs*	*Pectate lyase* (cleave pectin using a β-elimination mechanism)	*Cunninghamia lanceolata*	[36]
*XTH*	*Xyloglucan endotransglycosylases* (involved in the metabolism of xyloglucan)	*Cunninghamia lanceolata*	[36]
*GLUs*	*β-1,3-glucanase* (catalyzes the hydrolysis of β-1,3-glucosidic bonds existing in β-1,3-glucan)	*Cunninghamia lanceolata*	[36]
*BpEG*	*Endo-1,4-beta-glucanases* (cleave β-1,4-glycosidic bonds of cellulose)	*Cunninghamia lanceolata*	[36]
*miR397b*	(Inhibit *LAC4* and *LAC11*, involved in lignin polymerization)	*Cryptomeria fortunei* Hooibrenk	[49]
**Cell wall formation**			
*CESA*	*Cellulose synthase A* (the main enzyme that produces cellulose)	*Picea abies*	[10,13]
Monolignol biosynthesis genes	*CAD*, *COMT*, *CCoAOMT*, *CCR*, *C3H*, *C4H*, *4CL*, *CSE*, *PAL*, *HCT*	*Picea abies*	[13]
*Laccase*	(Lignin polymerization)	*Cunninghamia lanceolata*, *Picea abies*, *Cryptomeria fortunei* Hooibrenk	[13,36,49]
*Peroxidase*	(Lignin polymerization)	*Picea abies*	[13]
*AS2*	Homolog of *AtAS2* (regulates lignin deposition)	*Picea abies*	[13]
*MYB20*	Homolog of *AtMYB20* (positive regulator of SCW formation and lignification)	*Picea abies*	[13]
*MYB16*	Homolog of *AtMYB16* (regulation of cell shape and cuticle formation)	*Picea abies*	[13]
*MYB46*	Homolog of *AtMYB46* (master regulator of SCW formation)	*Larix kaempferi*	[51]
*MYB103*	Homolog of *AtMYB103* (Induce lignin formation)	*Larix kaempferi*	[51]
*miR319*	(Inhibits *TCP24*, a repressor of SCW formation)	*Ginkgo biloba* L.	[46]
**Programmed cell death**			
*MC5/9*	*Metacaspase 5/9* (cysteine protease that cleaves specifically after arginine or lysine residues)	*Picea abies*	[13]
*XCP1/2*	*Xylem Cysteine Protease 1/2* (cysteine protease involved in xylem tracheary element autolysis)	*Picea abies*	[13]
*BFN1*	*Bifunctional Nuclease1* (both RNase and DNase activities)	*Picea abies*	[13]
*RD21A*	*Responsive to Dehydration 21A proteinase* (both peptide ligase and protease activity)	*Picea abies*	[13]
*CEPs*	*Cysteine Endopeptidases* (hydrolysis of internal, alpha-peptide bonds in a polypeptide chain)	*Picea abies*	[13]
*XBCP3*	*Xylem Bark Cysteine Peptidase3* (xylem specific Cysteine peptidases)	*Picea abies*	[13]
*RNSI*	*class I RNAse* (RNA-degrading enzyme)	*Picea abies*	[13]
**Stress response (positive)**			
*TFL*	*Terminal Flower* (involved in growth cessation)	*Picea abies*	[10]
*BAM3*	β-amylase (involved in cold response)	*Picea abies*	[10]
*FTL2*	*Flowering Locus T/Terminal Flower1-Like 2* (involved in growth cessation)	*Picea abies*	[13]
*miR482*	(Involved in immune system)	*Ginkgo biloba* L.	[46]
*miR2118*	(Involved in fungal infection)	*Ginkgo biloba* L.	[46]
*miR2950*	(Involved in the plant immune system)	*Ginkgo biloba* L.	[46]
*miR5261*	(Involved in the Mg deficiency response)	*Ginkgo biloba* L.	[46]
**Plant growth**			
*MYB82*	(Involved in trichome development)	*Cryptomeria fortunei* Hooibrenk	[49]
*LaMYB84*	Homolog of *AtMYB84* (induce *LaCYCB1;3*)	*Larix kaempferi*	[51]
*MYB5*	(Involved in heat stress)	*Cryptomeria fortunei* Hooibrenk	[49]
*RDR*	*RNA-dependent RNA polymerase* (involved in posttranscriptional gene silencing)	*Picea abies*	[13]
*miR172*	(Promote vegetative induction)	*Cunninghamia lanceolata*	[71]
*miR156*	(Regulate the juvenile-to-adult transition)	*Cunninghamia lanceolata*	[71]
*miR396*	(Inhibit plant grown regulating factors (GRFs))	*Cryptomeria fortunei* Hooibrenk	[49]

## Data Availability

Not applicable.

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
