# Peer review of "Exploring the Seasonal Dynamics and Molecular Mechanism of Wood Formation in Gymnosperm Trees"

_ijms, 2023, doi:10.3390/ijms24108624_

Round 1

Reviewer 1 Report

1. Title  may be changed to "Exploring the Seasonal Dynamics and Molecular mechanism of Wood Formation in Gymnosperm Trees" to match the journal's content. 

2. Authors may write metabolites or proteins involved in wood formation in separate paragraphs.

3. One paragraph highlighting integrated interplay of climatic, hormonal, metabolites , genetics and epigenetics regulation may be given along with a figure.  Figure 1 is mostly highlighting the miRNAs and hormones.

Author Response

Comment #1: Title may be changed to "Exploring the Seasonal Dynamics and Molecular mechanism of Wood Formation in Gymnosperm Trees" to match the journal's content.

[Authors’ response 1] Thank you very much for your expert comment. We changed the title as you suggested, “Exploring the Seasonal Dynamics and Molecular mechanism of Wood Formation in Gymnosperm Trees” in this revised manuscript.

Comment #2: Authors may write metabolites or proteins involved in wood formation in separate paragraphs.

[Authors’ response 2] Thank you very much for your excellent comment. However, this aspect of wood formation (metabolites or proteins) is beyond the scope of this manuscript. If the opportunity arises, we would like to try it in the near future.

Comment #3: One paragraph highlighting integrated interplay of climatic, hormonal, metabolites , genetics and epigenetics regulation may be given along with a figure. Figure 1 is mostly highlighting the miRNAs and hormones.

[Authors’ response 3] Thanks for your comment. Actually, we tried to make the “Figure 1” to illustrate the entire process of wood formation in gymnosperm trees under climatic, hormonal, genetic and epigenetic (miRNA) regulations in two cell types; cambium and xylem. In addition, this manuscript (lines 380-382) summarizes the integrative interactions by stating that "Overall, seasonal regulation of wood formation in conifers is a complex and multi-faceted process that involves the interplay of various hormones, genes, miRNAs, and environmental cues ".

Reviewer 2 Report

This review summarizes and discusses the findings regarding the climatic, hormonal, genetic, and epigenetic (miRNA) regulation of wood formation (cell division,cell expansion,Cell wall formation,and Programmed cell death) in gymnosperm trees in response to seasonal changes.It is nessesaary and important to summarize these findings and progress with more and more good work being done in conifeer with several conifer genome sequenced.

Line 115, in addition to low temperature, short day-length is also key factor for some tree to entry into dormancy, because it can induce dormancy alone.

Line 13, delete “seasons”.

Line 53-55, the functional mechanisms of miRNA in plant are not described correctly and comprehensively.

In this MS, “the first” is used several times. Is is accurate? Please check it.

Line 65, change “Cryptomeria japonica  ”to “Cryptomeria japonica”.

Line 68, delete “.”.

Line 90, most of data are obtained from the work carried out in the Northern Hemisphere.Pay attention to the difference between the Northern Hemisphere and the Southern Hemisphere.

Line 93, change activation to activity.

Line 110, “It suggests that photosynthesis is more important than rainfall.” it is difficulty to understand or reach this conclusion. Please reconsider this sentence.

rest and quiescence are defined to be two different stages of dormancy based on the response of dormant tree to growth-promoting conditions in artificial condition. According to this, Rest occurs earlier than quiescence naturally, while quiescence occurs earlier than rest artificially [1]. In fact, it has been suggested that there exists a quiescence naturally before rest, and there also exists a quiescence naturally after rest [2-4].

Line 199, delete in.

Lieng 202, delete “the regulation of”.

In the part 5. Epigenetic regulation of seasonal wood formation, give a conclusion about this part.

Line 220, change “LaCYCB1-1” to LaCYCB1;1”.

Line 343-344, in some conifers, cambium cell division starts earlier than bud break, and IAA production occurs in the cambial regions, showing that “new shoots and leaves“ are only one of IAA sources, and stem is also a source.

Table 1,

Is LaMYB84 Involved in cork formation? If reference 51, this conclusion can not be found.

The picture showing the gene expression should be separated, the expression patterns of genes should be checked again and be sure that it is shown the same as reference. For example, is LaMYB20 expression high during reactivation? Please check.

References:

1. Espinosa-Ruiz, A.; Saxena, S.; Schmidt, J.; Mellerowicz, E.; Miskolczi, P.; Bakó, L.; Bhalerao, R.P. Differential stage-specific regulation of cyclin-dependent kinases during cambial dormancy in hybrid aspen. The Plant Journal 2004, 38, 603-615.

2. Li, W.F.; Ding, Q.; Chen, J.J.; Cui, K.M.; He, X.Q. Induction of PtoCDKB and PtoCYCB transcription by temperature during cambium reactivation in Populus tomentosa Carr. J EXP BOT 2009, 60, 2621-2630.

3. Mwange, K.N.; Wang, X.W.; Cui, K.M. Mechanism of dormancy in the buds and cambium of Eucommia ulmoides. Acta botanica sinica 2003, 45, 698-704.

please check the PDF I uploaded.

Author Response

Comments to the Author

This review summarizes and discusses the findings regarding the climatic, hormonal, genetic, and epigenetic (miRNA) regulation of wood formation (cell division, cell expansion, cell wall formation, and programmed cell death) in gymnosperm trees in response to seasonal changes. It is necessary and important to summarize these findings and progress with more and more good work being done in conifer with several conifer genome sequenced.

[Authors’ response] Thank you very much for your support and for taking the time to evaluate our manuscript (ijms-2367622). We really appreciate your favorable comments. Thanks again and have a great day!

Comment #1: Line 115, in addition to low temperature, short day-length is also key factor for some tree to entry into dormancy, because it can induce dormancy alone.

[Authors’ response 1] Thanks for your comment and we agree with you. Indeed, the short-day is also a key factor for dormancy induction that is described in this manuscript (lines 131-137). And the general description of the importance of the temperature and photoperiod in cambium activation was shown in lines 62-65.

Comment #2: Line 13, delete “seasons”.

[Authors’ response 2] Thanks for your comment. We deleted “seasons” (line 13) in this revised manuscript.

Comment #3: Line 53-55, the functional mechanisms of miRNA in plant are not described correctly and comprehensively.

[Authors’ response 3] Thanks for your comment. However, we tried to describe the general regulatory mechanism of miRNAs in this paragraph (lines 52-56).

Comment #4: In this MS, “the first” is used several times. Is is accurate? Please check it.

[Authors’ response 4] Thanks for your comment. There is a total of seven instances where "the first" is used. In this revised manuscript, we have made appropriate changes to avoid repetition and to be more precise: "earliest" (line 70), “beginning” (line 83, 95), "initial" (line 152), or just delete it (line 183).

Comment #5: Line 65, change “Cryptomeria japonica  ”to “Cryptomeria japonica”.

[Authors’ response 5] Thanks for your comment. We have changed to ‘Cryptomeria japonica‘ in this revised manuscript (line 64).

Comment #6: Line 68, delete “.”.

[Authors’ response 6] Thanks for your comment. We deleted “.” in this revised manuscript (line 67).

Comment #7: Line 90, most of data are obtained from the work carried out in the Northern Hemisphere. Pay attention to the difference between the Northern Hemisphere and the Southern Hemisphere.

[Authors’ response 7] Thanks for your comment. However, the aspect of ‘difference between the Northern Hemisphere and the Southern Hemisphere’ is beyond the scope of this manuscript. If the opportunity arises, we would like to try it in the near future.

Comment #8: Line 93, change “activation” to “activity”.

[Authors’ response 8] Thanks for your comment. We have changed it in this revised manuscript (line 93).

Comment #9: Line 110, “It suggests that photosynthesis is more important than rainfall.” it is difficulty to understand or reach this conclusion. Please reconsider this sentence.

[Authors’ response 9] Thanks for your comment. We agree, and have therefore removed that sentence from this revised manuscript.

Comment #10: “rest” and “quiescence” are defined to be two different stages of dormancy based on the response of dormant tree to growth-promoting conditions in artificial condition. According to this, Rest occurs earlier than quiescence naturally, while quiescence occurs earlier than rest artificially [1]. In fact, it has been suggested that there exists a quiescence naturally before rest, and there also exists a quiescence naturally after rest [2-4].

[Authors’ response 10] Thanks for your expert comment. In this manuscript (lines 119-124), we described the Rest occurs earlier than quiescence.

Comment #11: Line 199, delete “in”.

[Authors’ response 11] Thanks for your comment. We deleted “in” in this revised manuscript (line 200).

Comment #12: Lieng 202, delete “the regulation of”.

[Authors’ response 12] Thanks for your comment. We deleted “the regulation of” in this revised manuscript (line 203).

Comment #13: In the part “5. Epigenetic regulation of seasonal wood formation”, give a conclusion about this part.

[Authors’ response 13] Thanks for your comment. We included a conclusion as “In conclusion, miRNAs play a crucial role in regulating the different stages of cambium activity in gymnosperms, and their expression levels are tightly coordinated with hormonal and genetic regulation” in this revised manuscript (lines 349-351)

Comment #14: Line 220, change “LaCYCB1-1” to “LaCYCB1;1”.

[Authors’ response 14] Thanks for your comment. We changed the “LaCYCB1-1” to “LaCYCB1;1” in this revised manuscript (line 221).

Comment #15: Line 343-344, in some conifers, cambium cell division starts earlier than bud break, and IAA production occurs in the cambial regions, showing that “new shoots and leaves“ are only one of IAA sources, and stem is also a source.

[Authors’ response 15] Thanks for your comment. We modified the sentence to correct this by “The increase in the synthesis of auxins in new shoots and leaves and their transport to the vascular cambium, or production of auxins in the cambium, leads to the reactivation of this tissue, a critical component of secondary growth in plants” in this revised manuscript (lines 359-362).

Comment #16: Table 1, Is LaMYB84 Involved in cork formation? If reference 51, this conclusion can not be found.

[Authors’ response 16] Thanks for your excellent comment. It was our mistake and was corrected to “Homolog of AtMYB84 (induce LaCYCB1;3)” in Table 1 of this revised manuscript.

Comment #17: The picture showing the gene expression should be separated, the expression patterns of genes should be checked again and be sure that it is shown the same as reference. For example, is LaMYB20 expression high during reactivation? Please check.

[Authors’ response 17] Thanks for your comment. As you suggested, we separated the seasonal gene expression from Table 1 and made a new Figure 1 (see below).

And we checked the expression patterns of genes in Table 1 again and found no significant problems. In case of LaMYB20, we made a expression picture based on the following data from the reference 51.

Please check the uploaded pdf file. Thanks.

Reviewer 3 Report

The review article is devoted to the study of the influence of various factors on the formation of wood in coniferous plants.

A complex of external factors provides a change in the hormonal background, and through molecular interactions, allows the formation of early wood, and then passes into late wood.

The authors have done a large and interesting work covering the main molecular aspects of wood formation.

The work is well structured, the goals set are achieved.

There are some small comments that you can try to fix.

Part of the work is devoted to the fact that the authors consider miRNAs and their targets, as well as the role of this process in the formation of wood. It is necessary to emphasize in more detail what can affect on the expression of microRNAs themselves, in other words, what is the trigger for their formation at the molecular level.

Consider using more recent literature in your work.

In general, the review makes a very favorable impression.

Author Response

Comments to the Author

The review article is devoted to the study of the influence of various factors on the formation of wood in coniferous plants. A complex of external factors provides a change in the hormonal background, and through molecular interactions, allows the formation of early wood, and then passes into late wood. The authors have done a large and interesting work covering the main molecular aspects of wood formation. The work is well structured, the goals set are achieved.

[Authors’ response] Thank you very much for your support and for taking the time to evaluate our manuscript (ijms-2367622). We really appreciate your favorable comments. Thanks again and have a great day!

Comment #1: Part of the work is devoted to the fact that the authors consider miRNAs and their targets, as well as the role of this process in the formation of wood. It is necessary to emphasize in more detail what can affect on the expression of microRNAs themselves, in other words, what is the trigger for their formation at the molecular level.

[Authors’ response 1] Thanks for your excellent comment. However, the upstream regulators of the expression of the miRNAs described in this manuscript are still poorly understood, and studying them will be a really exciting science to understand this regulatory process.

Comment #2: Consider using more recent literature in your work.

[Authors’ response 2] Thanks for your comment. Actually, we are preparing research manuscript for this theme, which will be submitted soon.

Comment #3: In general, the review makes a very favorable impression.

[Authors’ response 3] Thanks. We appreciate it.

Round 2

Reviewer 2 Report

This MS has been improved based on the suggestion.

Comment #7: Line 90, most of data are obtained from the work carried out in the Northern Hemisphere. Pay attention to the difference between the Northern Hemisphere and the Southern Hemisphere.

[Authors’ response 7] Thanks for your comment. However, the aspect of ‘difference between the Northern Hemisphere and the Southern Hemisphere’ is beyond the scope of this manuscript. If the opportunity arises, we would like to try it in the near future. 

I know that the most work are done in the Northern Hemisphere, I have not suggested adding the work from Southern Hemisphere. What I want to express is that in the the same month the climate between the Northern Hemisphere and the Southern Hemisphere is different. So mid to late summer in Northern Hemisphere is from July to September, while in the Southern Hemisphere mid to late summer is from December to February.

Author Response

Response to Reviewer #2’s Comments

Reviewer #2

Comments to the Author

This MS has been improved based on the suggestion.

[Authors’ response] Thank you very much for your support and for taking the time to evaluate our manuscript (ijms-2367622). We really appreciate your favorable and helpful comments.

Comment #7: Line 90, most of data are obtained from the work carried out in the Northern Hemisphere. Pay attention to the difference between the Northern Hemisphere and the Southern Hemisphere.

[Authors’ response 7] Thanks for your comment. However, the aspect of ‘difference between the Northern Hemisphere and the Southern Hemisphere’ is beyond the scope of this manuscript. If the opportunity arises, we would like to try it in the near future.

Comment #: I know that the most work are done in the Northern Hemisphere, I have not suggested adding the work from Southern Hemisphere. What I want to express is that in the the same month the climate between the Northern Hemisphere and the Southern Hemisphere is different. So mid to late summer in Northern Hemisphere is from July to September, while in the Southern Hemisphere mid to late summer is from December to February.

[Authors’ response] Thank you for your kind suggestion. We revised in this 2nd revision by, “During mid to late summer (July to September) in Northern Hemisphere, day length is very long.” (lines 89-90).
